# Neoadjuvant Radiotherapy-Related Wound Morbidity in Soft Tissue Sarcoma: Perspectives for Radioprotective Agents

**DOI:** 10.3390/cancers12082258

**Published:** 2020-08-12

**Authors:** Cameron M. Callaghan, M. M. Hasibuzzaman, Samuel N. Rodman, Jessica E. Goetz, Kranti A. Mapuskar, Michael S. Petronek, Emily J. Steinbach, Benjamin J. Miller, Casey F. Pulliam, Mitchell C. Coleman, Varun V. Monga, Mohammed M. Milhem, Douglas R. Spitz, Bryan G. Allen

**Affiliations:** 1Free Radical and Radiation Biology Program, Department of Radiation Oncology, University of Iowa Hospitals and Clinics, Iowa City, IA 52422, USA; cameron-callaghan@uiowa.edu (C.M.C.); mm-hasibuzzaman@uiowa.edu (M.M.H.); sam-rodman@uiowa.edu (S.N.R.); krantiashok-mapuskar@uiowa.edu (K.A.M.); michael-petronek@uiowa.edu (M.S.P.); emily-steinbach@uiowa.edu (E.J.S.); douglas-spitz@uiowa.edu (D.R.S.); 2Holden Comprehensive Cancer Center and Department of Surgery, University of Iowa Hospitals and Clinics, Iowa City, IA 52422, USA; varun-monga@uiowa.edu (V.V.M.); mohammed-milhem@uiowa.edu (M.M.M.); 3Interdisciplinary Graduate Program in Human Toxicology, University of Iowa, Iowa City, IA 52422, USA; casey-pulliam@uiowa.edu; 4Department of Orthopedics and Rehabilitation, University of Iowa Hospitals and Clinics, Iowa City, IA 52422, USA; jessica-goetz@uiowa.edu (J.E.G.); benjamin-j-miller@uiowa.edu (B.J.M.); mitchell-coleman@uiowa.edu (M.C.C.)

**Keywords:** wound healing, soft tissue sarcoma, radiotherapy complications, radioprotective agents, neoadjuvant radiotherapy, limb preservation

## Abstract

Historically, patients with localized soft tissue sarcomas (STS) of the extremities would undergo limb amputation. It was subsequently determined that the addition of radiation therapy (RT) delivered prior to (neoadjuvant) or after (adjuvant) a limb-sparing surgical resection yielded equivalent survival outcomes to amputation in appropriate patients. Generally, neoadjuvant radiation offers decreased volume and dose of high-intensity radiation to normal tissue and increased chance of achieving negative surgical margins—but also increases wound healing complications when compared to adjuvant radiotherapy. This review elaborates on the current neoadjuvant/adjuvant RT approaches, wound healing complications in STS, and the potential application of novel radioprotective agents to minimize radiation-induced normal tissue toxicity.

## 1. Introduction

Soft tissue sarcomas (STS) are a relatively rare group of malignancies with multiple histological subtypes [1]. The majority of STS originate from the extremities (46% lower, 13% upper) [2,3,4,5,6,7,8,9,10,11,12,13,14,15,16,17,18,19,20,21,22,23,24,25,26,27,28,29,30,31,32,33,34,35,36,37,38,39,40,41,42,43], but may arise in any region including the torso/trunk (18%) [21,35,39,44], retroperitoneum (13%) [45,46,47,48,49,50], or head and neck (9%) [51,52]. Because STS commonly presents as a painless enlarging mass, diagnosis is often delayed until tumors become large in volume, often abutting critical nerves and vessels [53].

STS can be locally infiltrative with microscopic tumor deposits extending up to 4 cm beyond the primary tumor [28,54], limiting the ability of surgeons to preserve limbs without risking microscopic residual disease or positive margins. Amputation was therefore the primary treatment modality for STS of the extremities until limb-sparing surgeries combined with radiotherapy (RT) showed similar outcomes [55]. Randomized prospective trials and retrospective studies demonstrated similar local control and overall survival rates between limb-sparing surgery combined with RT compared to amputation [43,56], as well as the importance of including RT for successful limb-sparing resection [4,57]. The addition of RT is thought to eliminate microscopic residual tumor cells located around the gross tumor. Any additional RT after resection is based on the patients risk for local recurrence.

Modern limb-sparing surgery aims to achieve similar local tumor control and survival outcomes compared to amputation, while preserving as much long-term limb function as possible. While RT improves survival and local control outcomes, it also increases the risks of acute sequelae including acute wound complications and radiation dermatitis [30,36,51,58,59], as well as late toxicities of fibrosis, necrosis, edema, pathologic fractures, and long term decrease in limb function [11,30,36,51,52,58,59].

RT in STS can be delivered pre-operatively (neoadjuvant), intraoperatively (IORT), or post-operatively (adjuvant) via external beam RT (EBRT) or with brachytherapy (BT) using radioactive isotopes. While the choice of neoadjuvant vs. adjuvant radiotherapy is always considered on an individualized patient basis, most studies have demonstrated equivalent disease control, but significant differences in toxicity profiles between these two approaches. In general, neoadjuvant RT is associated with more acute wound complications [51,58] while adjuvant RT is associated with higher rates of late toxicities and decreased limb function [52,58]. Current guidelines slightly favor neoadjuvant RT because of reduced radiation dose and reduced radiation volumes thereby reducing the cumulative exposure of normal tissues to RT.

Despite its benefits, neoadjuvant RT is known to impair wound healing and cause significant long-term morbidity (Figure 1). Even with improved image guidance and modern RT delivery techniques, approximately one-third of STS patients receiving neoadjuvant RT still experience wound healing complications. Understanding the mechanisms of radiotherapy-related normal tissue toxicity which lead to impaired wound healing may result in the identification of pathways and therapeutic targets for mitigation of radiation-induced injury, to protect normal tissue. In this review, we will discuss the application of radiotherapy in STS and associated problems with wound healing, cellular and molecular events dysregulated by RT, and historical and novel radioprotective agents currently under investigation for the prevention radiation-induced normal tissue toxicity.

## 2. Radiotherapy in Soft Tissue Sarcoma (STS)

The discovery of similar local control and overall survival outcomes between amputation and limb-sparing surgery combined with RT have led to this approach becoming the standard of care for STS of the extremity [60]. Since the initial introduction of RT, there have been tremendous advancements in image guidance, radiation delivery techniques/modalities, and clinical regimens for combining radiation with surgery and/or chemotherapy. We will examine radiotherapy in STS by radiation modality, clinical regimen (neoadjuvant vs. adjuvant), and anatomic disease site with associated toxicity outcomes.

### 2.1. Radiotherapy Modalities

While 3D Conformal Radiation Therapy (3D-CRT) is the more traditional method of planning External Beam Radiation Therapy (EBRT) treatments, for many tumors Intensity Modulated Radiation Therapy (IMRT) provides the best dose conformity to the target area while reducing toxicity to normal structures [30,36,45,48]. Brachytherapy (BT) utilizes radioactive isotopes to deliver RT from within the target volume, while Intraoperative radiotherapy (IORT) refers to the delivery of dose at the time of surgery (which can be achieved via either brachytherapy or linear accelerators).

#### 2.1.1. External Beam Radiation Therapy (EBRT)

Target coverage and protection of normal tissues appear to be superior for IMRT compared with 3D-CRT in STS of the extremity [17,60] and the retroperitoneum [45,48]. Retrospective studies have demonstrated that patients treated with IMRT have lower rates of local recurrence compared with 3D-CRT (7.6% IMRT vs. 15.1% 3D-CRT; *p* = 0.02) [19,36,61] or BT (8% IMRT vs. 19% BT; *p* = 0.04) [24]. Though there has never been a prospective trial randomizing patients to IMRT vs. either 3D-CRT or BT, IMRT has been associated with lower rates of wound complications compared with historical 3D-CRT results (30.5 vs. 43%, respectively) [11,30], but higher rates compared with BT (19% IMRT vs. 11% BT) [24] though neither of these results were statistically significant. IMRT has some evidence of lower rates of femoral fracture [62].

#### 2.1.2. Brachytherapy (BT)

Brachytherapy can be used for Intraoperative Radiotherapy (IORT; discussed next section) [46,63,64] or to deliver adjuvant radiation for low-risk/re-irradiation cases as a monotherapy [5,57], or as a boost in combination with EBRT for high-risk cases, or in cases in which the target volume cannot easily be covered by BT alone [5,65]. Brachytherapy can also shorten total treatment time for patients (e.g., 4–5 days for 45 Gy via LDR brachytherapy vs. 5–6 weeks for IMRT) [66,67].

In one of the few prospective trials with randomization with respect to radiation, adjuvant BT monotherapy improved 5-year local control for patients with high-grade STS of the extremities or superficial trunk as compared with no BT (89% BT vs. 66% no BT) [57]. However, at least one analysis of a prospective trial (not randomized with respect to radiation modalities), demonstrated an inferior 5-year local control for BT monotherapy as compared with IMRT (81% BT vs. 92% IMRT, *p* = 0.04), with a non-significant difference in 5-year overall survival (73% BT vs. 62% IMRT, *p* = 0.1) [24]. Other retrospective studies have shown mixed results when comparing EBRT vs. BT vs. EBRT + BT boost [5,6,68].

One retrospective study of adjuvant LDR BT monotherapy had lower rates of wound complications compared with historical EBRT results with 5-year actuarial rates of wound complications requiring reoperation, bone fracture, and grade ≥3 nerve damage of 12, 3, and 5%, respectively [6] with similar findings in studies of HDR-BT [69]. One study evaluating HDR-BT monotherapy vs. EBRT vs. EBRT + HDR-BT boost noted higher incidents of seroma/hematoma and deep infection in BT cohorts, whereas EBRT cohorts had greater incidents of chronic edema, fibrosis, and radiation dermatitis [70].

When BT is used as a boost to EBRT, one non-randomized study demonstrated National Cancer Institute (NCI) grade 2–4 wound healing complications of 40 and 18% for LDR and HDR brachytherapy, respectively (though this was not significant at *p* = 0.14). In this study, complications with LDR were correlated with suboptimal implant geometry, while for HDR they were correlated with dose per fraction, total dose, and total biological equivalent dose [65]. Other studies of HDR-BT combined with EBRT have confirmed similar rates of acute and late toxicity [71], with the volume of tissue receiving >150% of the prescription dose being a possible predictor of toxicity, especially in the lower extremities [72].

#### 2.1.3. Intraoperative Radiotherapy (IORT)

Intraoperative RT (IORT) delivered via brachytherapy [24,46,64] or via an electron beam using specialized linear accelerators [15,34,41,73,74] is almost always combined with EBRT as a means of boosting especially high-risk volumes in the extremity [15,16,33,34,41,73] or retroperitoneum [74,75,76,77,78]. IORT allows for moving at-risk tissues away from the radiation field or blocking off organs at risk using lead shields. IORT requires smaller treatment volumes and a lower total dose (~10–20 Gy) but in a higher dose per fraction.

IORT in combination with EBRT provides excellent local control in STS of the extremities [15,16,33,34,41,56,73,79] and the retroperitoneum [46,74,75,76,77,78] with high rates of good functional outcomes and limb preservation. As IORT is almost always paired with some form of EBRT it is difficult to assess which toxicities can be attributed to IORT as opposed to EBRT. Moderate to severe acute toxicities (mostly radiation dermatitis) have ranged from 1–24% [15,34,41] with acute wound complications, including the need for revision surgery, ranging from 5–36% [34,78,80,81]. One study noted that the rate of wound complications varied significantly based on whether IORT was paired with neoadjuvant vs. adjuvant EBRT (36 vs. 15%, respectively) [80]. Late toxicities including fractures, neuropathy, and fibrosis, ranged from 10–20% on long term follow up [15,34,41,78,81] with one study noting that a rate of 12% for all grades of neuropathy, which increased to 25% in patients who had a major nerve passing through the high dose IORT field [81].

### 2.2. Clinical Regimen-Neoadjuvant, Adjuvant, and Combined Modality Radiotherapy

#### 2.2.1. Neoadjuvant Radiotherapy

A typical regimen of neoadjuvant RT in STS consists of 50 Gy delivered in 1.8–2.0 Gy once-daily fractions over 5–6 weeks [82], providing a lower cumulative dose and smaller treatment fields, which are achieved by better target delineation and image guidance [37]. Other aims of neoadjuvant RT include sterilization of microscopic disease on the edge of the tumor and induction of a pseudocapsule around the primary tumor to aid in obtaining negative margins during resection [83,84,85]. Pseudocapsule generation may also allow for preservation of critical structures, improved post-operative functional status, and decreased risk of seeding during resection.

Despite several advantages of neoadjuvant RT, a higher risk of postoperative wound complications remains a substantial challenge. Several studies have reported higher rates of wound complications in neoadjuvant radiotherapy [51,58,86,87], but lower rates of chronic side effects including edema, fibrosis, fracture, and joint stiffness compared with adjuvant RT [11,52,58,88] (Table 1). One study comparing neoadjuvant (50 Gy/25 fractions) to adjuvant RT (66 Gy/33 fractions) demonstrated higher incidence of major wound complications (35 vs. 17% respectively) [58]. The time from completion of neoadjuvant RT to surgery may also influence the rate of acute wound complications with one study suggesting 3-6 weeks as optimal [38] while longer delays may lead to late radiation fibrosis and increased surgical complications [82].

#### 2.2.2. Adjuvant Radiotherapy

Adjuvant RT is typically delivered via EBRT to a total dose of 60–66 Gy in 1.8–2 Gy fractions usually 2–4 months after surgical resection to eliminate microscopic residual disease [4,82] but can also be delivered via brachytherapy as discussed in the above section [57]. Compared with neoadjuvant RT, adjuvant radiation allows for better staging of tumor grade and appropriate surgical margins to be achieved without any impact of prior RT on tumor [13]. Additionally, adjuvant RT has reduced the incidence of acute wound complications which require additional surgical procedures [30,51,58,59]. Several studies have confirmed lower acute radiation toxicity in the adjuvant setting (Table 1). In a prospective, randomized trial, Davis et al. showed a significantly higher incidence of fibrosis (48.2% vs. 31.5%), edema (23.2% vs. 15.1%) and fracture (23.2% vs. 17.8%) in adjuvant RT compared with neoadjuvant RT, respectively (all, *p* < 0.05) [11]. These late-stage complications may be related to increased total radiation dose (50 Gy in neoadjuvant vs. 60–66 Gy adjuvant) and larger treatment fields necessitated by surgical resection [52]. Most clinical studies regarding the timing of surgery and RT in STS use local control and wound morbidity as primary endpoints, but few studies have attempted to explain the mechanism of RT-induced normal tissue injury or wound complications in STS [11,30,36,51,52,58,59].

#### 2.2.3. Intraoperative and Adjuvant Boosts

Radiation field boosts in STS are generally delivered to smaller volumes consider to be at higher risk for recurrence. Whether EBRT is delivered neoadjuvant or adjuvantly, the boost may be delivered via IORT, adjuvant brachytherapy, or additional EBRT fractions. Recommended boost RT doses vary following neoadjuvant RT, and are determined by RT modality and surgical margin status (16–18 Gy for microscopically positive and 20–26 Gy for grossly positive margins when using an EBRT boost; 16–26 Gy LDR or 14–24 Gy HDR for brachytherapy boost; and typically 10–12.5 Gy for microscopically positive, and ~15 Gy for grossly positive margins for an IORT boost) [16,66].

If there is an indication for RT boost prior to or during surgery, IORT may be delivered (~10–16 Gy) or catheters placed for an adjuvant BT boost (~16–20 Gy LDR or HDR equivalent for positive margins). If an IORT or BT boost were delivered, post-operative EBRT would usually be initiated 3–8 weeks after surgery. An alternative to BT or IORT boost following neoadjuvant RT is to boost tissues via EBRT (10–16 Gy delivered over 5 to 8 fractions) in the post-op setting. However, some studies demonstrated no local control benefit of an adjuvant boost for positive surgical margins [89,90].

### 2.3. Anatomic Disease Site

STS are classified into different staging groups by the American Joint Committee on Cancer (AJCC) 8th edition into categories of “Head and Neck”, “Trunk and Extremities”, “Abdomen and Thoracic Visceral Organs”, and “Retroperitoneum”, while the National Comprehensive Cancer Network (NCCN) guidelines split STS anatomically into groups of ‘Extremity/Body wall/Head and Neck’ and “Retroperitoneal/Intraabdominal”, with Rhabdomyosarcoma, Desmoid, and Gastrointestinal Stromal Tumors being treated as separate entities.

#### 2.3.1. Extremity, Head and Neck, and Superficial Trunk

In AJCC stage IA-IB, low-grade disease-RT is typically reserved for cases in which appropriate margins were not achieved during surgery, though re-resection (if feasible) or observation (for IA) are also management options. In patients with resectable, stage II-III disease, which would be likely to have acceptable functional outcomes after surgery-RT can be delivered neoadjuvantly or adjuvantly, possibly with chemotherapy for stage III disease. Chemotherapy is not the standard of care in STS. There is currently no Category 1 evidence to suggest an overall survival benefit by treating STS patients with chemotherapy alone or in combination with RT in the non-metastatic, locally advanced setting. However, chemotherapy has been employed in select patients and can be considered in unresectable stage II-III disease or cases in which acceptable functional outcomes would not be expected after surgery. Management options in these cases include RT, chemotherapy, chemoradiation, or amputation (extremity). Once the patient receives RT and/or chemotherapy, they can then be re-evaluated to assess whether they have become a suitable candidate for surgery.

In general, there are studies that are specific to STS of the extremities [40,41,42,43], but most that include STS of the Head and Neck or Trunk/Torso/Body wall are combined with extremity STS cases [51,52,91,92,93]. There are indications that extremity STS (especially lower extremities) have higher rates of wound complications as well as unique considerations/options involving amputation vs. limb preservation. Additionally, certain RT modalities are used more or less frequently based on anatomic location (e.g., the use of IORT is relatively common in the retroperitoneum, whereas adjuvant BT in the upper abdomen is not recommended) [66].

#### 2.3.2. Retroperitoneal/Intra-Abdominal

For retroperitoneal STS, adjuvant RT is typically not administered for patients that have negative or microscopically positive margins unless local recurrence would cause significant morbidity [50]. If a patient underwent neoadjuvant RT and ultimately had microscopically positive margins, a 10–16 Gy boost may be considered per current NCCN guidelines [94]. Studies typically separate retroperitoneal STS from those of the Superficial Trunk/Head and Neck/Extremities, with or without “Intra-abdominal” or “Visceral Organ” STSs included.

## 3. Normal Physiology of Wound Healing

Wound healing involves four distinct but interconnected phases; early hemostasis, inflammation, late proliferation, and remodeling [95]. Impaired wound healing occurs when there are extended inflammatory and proliferative phases resulting in pathological fibrosis, scarring, and non-healing ulcers [96,97].

### 3.1. Early Homeostasis and Inflammation

The initial response after the injury is mediated by platelets which release platelet-derived growth factor (PDGF) and Transforming Growth Factors alpha and beta (TGF-α and TGF-β) to recruit neutrophils, macrophages, and leucocytes which go on to remove wound debris, secrete anti-microbial factors, pro-inflammatory cytokines (IL-1, IL-6, IL-8, and TNF-α), and growth factors (PDGF, TGF-β, TGF-α, IGF-1, and FGF) which ultimately recruit fibroblasts and epithelial cells to the site of the wound [98,99,100].

### 3.2. Late Proliferation

The late proliferative phase is characterized by the laying down of collagen fibers, proteoglycans, and other matrix components orchestrated by fibroblasts, keratinocytes, and endothelial cells [99]. Fibroblasts activated by platelets and macrophages migrate into the wound and bind matrix components via their integrin receptor, change morphology, and begin secreting matrix metalloproteinases (MMPs) which clear a path for the fibroblasts movement from the extracellular matrix (ECM) into the wound site [100]. Many MMPs are involved, most notably collagenase (MMP-1), gelatinases (MMP-2 and MMP-9), and stromelysin (MMP-3) [101]. After migration through the collagen matrix, fibroblasts begin to proliferate and synthesize granulation tissue components including collagen, elastin, and proteoglycans. PDGF and TGF-β secreted by platelets and other fibroblasts have been shown to be key regulatory growth factors regulating fibroblast activity [100]. Angiogenesis accompanies the granulation phase and is stimulated by soluble mediators (bFGF, TGF-β, and VEGF) secreted by epithelial cells, fibroblasts, vascular endothelial cells, and macrophages [102].

### 3.3. Remodeling

Formation of new blood vessels creates a favorable microenvironment for the migration, proliferation, and differentiation of dermal and epidermal cells leading to wound re-epithelialization and matrix deposition. Granulation tissue matures into scar tissue and the collagen matrix ultimately is replaced by ECM in the remodeling phase. Wound contraction and matrix remodeling are mediated mainly by fibroblast motility and simultaneous matrix reorganization. This is facilitated by α-smooth muscle actin (SMA) secreted by fibroblasts which differentiate into myofibroblasts in response to TGF-α [103]. The balance between protease secretion and inhibition is regulated by MMPs and serine proteases, which are in turn regulated by Tissue Inhibitor of Metalloproteinases (TIMPs) for MMPs and α1 protease inhibitor and α2 macroglobulin for serine proteases respectively [100]. It has been demonstrated that apoptosis of fibroblasts and myofibroblasts at the end of the normal wound healing is important for the prevention of pathologic scars and keloid formation [97,100].

## 4. Wound Pathology after Radiation Treatment

Wound complications from neoadjuvant RT as part of STS treatment may also depend on surgery-related injuries from the surgical approach and patient demographics such as age, sex, diabetes, and other chronic comorbidities. RT causes increased production of Reactive Oxygen Species (ROS) by radiolysis of water, which damages cellular DNA, proteins, and lipid molecules resulting in the injury of the stratum basale and ultimately delays wound healing [104,105]. ROS, including hydrogen peroxide (H_2_O_2_) and superoxide (O_2_^•−^), also serve as intracellular messengers that regulate key steps of wound healing. Initially, high levels of ROS lead to vasoconstriction and thrombus formation, platelet aggregation, inflammatory cell recruitment, and killing of microorganisms across the wound area [106]. Persistent oxidative stress impairs wound healing by inducing apoptosis, senescence, and by prolonging the inflammatory phase. Many of the cytokines involved in the formation/persistence of a chronic wound (including PDGF, TNF, and ILs) also induce the production of ROS and Reactive Nitrogen Species (RNS), having downstream effects on a wide range of processes including migration, fibrosis, aging, and senescence [107,108,109,110,111].

Several molecular concepts have been derived from human and animal studies regarding the mechanism of normal tissue injury following radiation exposure. One notable finding is that in general, radiated cells have reduced mitogenic capacity. Unrepaired or partially repaired DNA damage caused by ionizing radiation may result in mitotic or clonogenic cell death and apoptosis. In a murine model, Franklin et al. showed that radiation slows down dermal filling, but maintains a normal rate of re-epithelialization, suggesting irradiated cells have an impaired ability to proliferate, but maintain the ability to migrate [105]. Several in vitro studies have demonstrated impaired DNA synthesis in chronic wounds compared with acute wounds [112,113]. Other studies have found the associations between radiation-induced inhibition of cellular proliferation, cell cycle arrest, and apoptosis, with delayed granulation tissue formation [114].

Cells can also lose their proliferative capacity following radiation and the associated DNA damage can lead to a permanent-growth arrest (senescence) without undergoing cell death. These senescent cells have a secretory phenotype which may produce a large number of inflammatory cytokines, chemokines, and growth factors. Senescent fibroblasts secret MMP-2, 3, and 9 which may delay fibrosis [115]. Furthermore, senescent keratinocytes secrete mapsin, an anti-angiogenic factor that can be detrimental to wound repair [116]. ROS-mediated senescence is also believed to induce fibroblast proliferation and keloid formation [117].

Another major concept of RT-induced normal tissue injury is prolonged inflammation (Figure 2). Although early inflammation is favorable for wound healing, prolonged inflammation can activate a series of inflammatory cytokines, chemokines, and growth factors leading to fibrosis and excessive scarring [118]. Within a few hours of radiation delivery, various pro-inflammatory cytokines (IL-1, IL-8, IL-3, IL-6, TNF-α), chemokines (eotaxin, CCR receptor), and adhesive molecules (ICAM-1, V-CAM, E-selectin) are released, leading to events that prolong inflammation, uncontrolled matrix accumulation, and fibrosis [119,120]. Major cytokines involved in chronic and acute inflammation are IL-1, IL-8, IFN-γ, and TNF-α. Several animal models have found higher levels of IL-1, TNF-α, and IFN-γ in irradiated wounds [120,121]. These cytokines may activate fibroblasts and keratinocytes, as well as MMPs, which play a critical role in the late stages of wound healing. As MMPs are prerequisite for fibroblast activation and migration, increased levels of MMPs in the remodeling phase leads to enhanced migration of fibroblasts and elevated production of matrix components, resulting in fibrosis [34]. Another important mediator produced by macrophages and fibroblasts is nitric oxide (NO), which accelerates collagen formation, cell proliferation, and wound contraction [122]. It has been shown that RT leads to decreased levels of NO in irradiated wounds [123].

Irradiation may induce microvascular damage and local tissue hypoxia. Ionizing radiation thickens the basement membrane and increases the permeability of blood vessels generating edema in the surrounding tissue. This impairment of metabolic and gas exchange leads to hypoxia. Although transient hypoxia is beneficial due to its activation of the initial inflammatory response, prolonged hypoxia resulting from progressive endothelial thickening may be detrimental. Hypoxia generates ROS, activates pro-fibrotic cytokines (TGF-β), and promotes collagen formation [124,125,126]. Moreover, damage to the vasculature and concomitant release of vasoactive cytokines (IL-1, IL-6, and TNF-α), growth factors (PDGF), and downregulation of NO synthase and thrombomodulin can activate a series of events leading to fibrosis [127,128]. Thus, the toxicities following tissue irradiation can be characterized by persistent and uncontrolled inflammation, elevated levels of proteases including MMPs, and inflammatory cells. A combination of these factors ultimately results in the inhibition of cellular proliferation and migration, impaired matrix remodeling, imbalance in pro-inflammatory and anti-inflammatory cycles, and accumulation of necrotic tissue due to tissue hypoxia.

## 5. Musculoskeletal Injury from STS Radiotherapy

The treatment of STS with RT often results in dose to articular joints and other musculoskeletal tissues that exhibit specific injuries and related wound healing responses as a result of radiation therapy. A retrospective study of 145 patients receiving radiation with and without chemotherapy for extremity STS showed that after more than one year following treatment 6% suffered bone fracture, 20% developed joint contracture, 19% suffered edema, 20% experienced muscle weakness and 57% developed an accumulation of fibrotic tissue in and around the wound site [2]. These secondary musculoskeletal effects from radiation therapy can have a substantial impact on patients’ quality of life.

Basic research into joint contracture after radiation therapy has concentrated on the characterization of incidence rates after specific doses. Initial work in murine models demonstrated a dose-dependent joint contracture with a single dose of radiation between 20–80 Gy on the hind limb, with skin contracture playing a dominant role in fibrosis from 20–30 Gy, while above this range, deeper tissues were responsible for contracture [129]. When fractionated doses were delivered to mice, removal of skin and muscle did not completely account for the total contracture shown [130]. When these models included a fibrosarcoma in conjunction with the radiation treatment, joint contracture was greater in these mice relative to controls, suggesting that tumors have a detrimental effect on wound healing and normal tissue injury [131]. Radiation therapy has also been combined with hyperthermia treatment to examine the effects of this combination on normal tissue injury in the same murine model. This combination demonstrated significant exacerbation of limb contracture when compared with radiation alone at temperatures ranging from 43.0–43.5° Celsius [132].

An additional mid- to long-term complication that can substantially impact the quality-of-life of sarcoma survivors is a radiation-associated fracture. The most recent estimates place the risk of femur fracture at 7%, after treatment with 50–66 Gy, with a median time to fracture of 2 years [62]. Radiation treatment can alter the mechanical strength of the bone [133], making it more prone to fracture after a subclinical force or progress to a complete fracture after the development of a stress reaction. Not only does radiation make the bone more brittle, but it also decreases its capacity to heal. Resection of a deep soft tissue sarcoma often requires removal of the surrounding periosteum to ensure a negative margin. This compromises the external osseous blood supply, further limiting regenerative capacity after a bone injury. Even with operative fixation, the rate of non-union is reported to be over 80% [134] and many cases must be definitively managed with resection of the nonunion bone and implantation of a metallic endoprosthesis or amputation.

## 6. Radioprotective Agents

The acute toxicities of RT are mostly due to radiogenic cell killing, whereas late RT associated toxicities are due to prolonged inflammation and dysregulation of growth factors involved in wound healing [119]. One approach to decrease acute and chronic RT induced toxicity are radioprotective agents to counterbalance the radiation damage (Figure 3). However, if the radioprotective agent does not exploit a unique biological feature which differentiates normal and malignant cells, it runs the risk of protecting tumor cells from radiation as well as healthy tissue. This could potentially still open a therapeutic window so long as the agent is a more potent radioprotector of healthy tissues than malignant cells but would lead to the additional complication of determining the new optimal radiation dose to be delivered alongside the radioprotective agent. Fortunately, some agents discussed below demonstrate simultaneous radioprotection in normal tissue and radiosensitization of cancer cells. With a better understanding of the critical steps of wound healing impaired by radiotherapy, the development of radiation protectors and mitigators may help decrease these toxicities and improve outcomes.

### 6.1. Amifostine

Amifostine is an antioxidant and radioprotector that works by scavenging hydroxyl radicals (HO^•^) [135]. In a phase III clinical trial of squamous head and neck cancer patients, amifostine was found to have protective effects on acute and delayed xerostomia but not severe oral mucositis [136] and has been studied broadly in clinical trials with RT with or without concurrent chemotherapy [137]. Based on these findings, amifostine is currently recommended by the American Society of Clinical Oncology (ASCO) for the prevention of xerostomia during fractionated radiotherapy [138]. In a study by Margulies et al., amifostine was shown to protect normal chondrocytes and osteoblasts from radiation damage but failed to protect bone marrow during the treatment of Ewing’s sarcoma or rhabdomyosarcoma [139]. Amifostine has even shown evidence of increasing the efficacy of RT in the context of a rhabdomyosarcoma rat model [140]. Several recent clinical trials have been/are currently being conducted to investigate the ability of amifostine to reduce RT side effects in the rectum as part of prostate cancer treatment (NCT00040365) [141], reduce head and neck side effects in the treatment of lymphoma (NCT00136474) [142], or for reducing neuropathy from paclitaxel in a variety of settings (NCT00078845) [143]. The results of these studies may suggest whether amifostine should be investigated further as a radioprotective agent in STS treatment.

### 6.2. Nitroxides

Nitroxides are stable free radical compounds that protect against ROS-mediated cellular injury by direct scavenging of free radicals and inhibiting Fenton reactions. Nitroxides are a low molecular weight compound that can act as superoxide dismutase mimics and play a role in the inhibition of lipid peroxidation [144]. Due to its ROS scavenging properties, nitroxides have been investigated as a radioprotector in chemotherapy and RT-associated oral mucositis, nephrotoxicity, and ototoxicity [145,146,147,148]. In addition to nitroxides protection against RT damage in normal tissue, high concentrations of nitroxides can enhance radiation-induced oxidative damage in cancer cells through the release of iron and activation of the Fenton reaction [149]. Nitroxides may have tissue selectivity depending on the different oxygenation states in normal vs. tumor tissue. In the oxidized state, the nitroxide radical acts as a free radical, a SOD mimic, and a radioprotector. In the reduced state, the hydroxylamine is a non-radical compound with antioxidant properties that does not act as a radioprotector [150]. For example, tempol remains in its oxidized state due to normal tissue being well oxygenated but reduced in hypoxic tumor tissue [151]. In a preclinical study in vivo, tempol was found to protect bone marrow from lethal radiation doses (36.7 Gy) without affecting tumor control [152]. Additional studies established tempol’s protective effects against lethal total body radiation in a murine model [148,153]. In another mouse model combining fractionated RT with a mitochondrial-targeted tempol (GS-nitroxide, JP4-039), JP4-039 was able to protect bones from injury [147]. Long-term side effects of sarcoma treatment include bone fracture from both neoadjuvant and adjuvant RT and provide a greater rationale for further investigations of the application of nitroxides to mitigate radiation-associated toxicity in sarcoma treatment.

### 6.3. Pharmacological Ascorbate

Intravenous pharmacological ascorbate (intravenous infusions of vitamin C resulting in plasma ascorbate concentrations ≥20 mM; P-AscH-) has been shown to act as a radiosensitizer in cancer cells and as a radioprotector in normal cells [154,155,156]. Due to fundamental differences in H_2_O_2_ and redox-active iron metabolism, pharmacological ascorbate acts as an antioxidant in normal tissues and pro-oxidant in cancer cells [155]. In an in vitro model of fibrosarcoma (HT-1080) and liposarcoma (SW872), pharmacological ascorbate sensitized sarcoma cells to both chemotherapy and radiation through an intracellular labile iron-dependent mechanism and H_2_O_2_. These results were recapitulated in vivo using an orthotopic sarcoma model in nude athymic mice [157]. In an in vivo model of radiogenic wound healing, pharmacological ascorbate was found to induce a significant acceleration in healing of radiogenic ulcers in mice treated with 10, 16, and 20 Gy of radiation. Following RT, mice received a full-thickness skin wound by biopsy puncture. Earlier wound closure and increased synthesis of collagen were observed in mice treated with ascorbic acid as compared with the control group [158]. A phase Ib/II clinical trial is currently underway combining pharmacological ascorbate with neoadjuvant RT in the treatment of locally advanced, resectable, high-grade STS (NCT03508726) [159]. Thus, pharmacological ascorbate poses a clinically relevant advancement for the treatment of sarcoma patients to sensitize cancer cells to radiation while simultaneously protecting normal cells from radiation damage.

### 6.4. Superoxide Dismutase Mimetics

Exogenous superoxide dismutase (SOD) mimetics are a class of compounds that can act as radiosensitizers in cancer cells and radioprotectors in normal cells [160]. CuZnSOD and Mn-SOD are two forms of SOD that can reverse radiation-induced fibrosis [160]. Several studies have found that CuZnSOD results in repression of TGFβ1, phenotypic reversion of myofibroblasts, and increased collagen degradation due to reduced TIMP activity [161,162,163]. GC4419 (previously known as M40419) is a small molecular weight mitochondria-specific SOD mimetic with a dismutation of superoxide rate constant of 1 × 10^7^/mol/s; this compares favorably with the native MnSOD enzyme at 1.2 × 10^9^/mol/s [164,165]. Previous research showed that older fibroblasts have less migration ability compared with younger dermal fibroblasts, and GC4419 can restore the migration ability of the older dermal fibroblasts [160]. In a phase Ib/IIa clinical trial in patients with head and neck cancer treated with ≥60 Gy of radiation with concurrent cisplatin, GC4419 was found to have protective effects against oral mucositis [166]. With regards to radiation dermatitis, Doctrow et al. showed that synthetic SOD/catalase mimetic, EUK-207, could mitigate radiation dermatitis, decrease expression of oxidative stress markers, and enhance wound healing [167]. These studies indicate that exogenous SOD mimetics may be an effective therapy to minimize radiation-induced STS wound healing.

### 6.5. Other Radioprotectors

Additional antioxidants including glutathione, lipoic acid, and vitamins A and E have been studied in preclinical and clinical models. Although many of these antioxidants were tested for the treatment of different RT-induce side effects, including xerostomia [168], oral mucositis [169], and pulmonary fibrosis [170], a major concern of these radioprotectants is the non-selective scavenging of free radicals. Radioprotection of tumor cells has been observed clinically, highlighting the need for simultaneous investigation of tumor efficacy and radioprotection of normal tissues for all agents prior to clinical implementation.

#### 6.5.1. Vitamin E

The vitamin E family of antioxidants consisting of tocopherols and tocotrienols. Multiple studies have reported normal tissue radioprotection and antitumor activity with the use of both tocopherols and tocotrienols [171]. Although studies regarding the use of vitamin E in diabetic wound healing and other chronic wounds have been reported, preclinical and clinical studies on the effect of RT induced changes in wound healing are sparse [172,173]. However, the mechanism of vitamin E indicates that it could be used to treat wounds following RT, including those seen in the treatment of STS. For example, tocotrienols scavenge or catalytically inactivate free radicals through the expression of antioxidant enzymes, including SOD and glutathione peroxidase [174,175]. Additionally, tocotrienols contain anti-inflammatory effects mediated by the expression of anti-inflammatory molecules and suppression of proinflammatory cytokines through NF-κB signaling [176,177]. Thus, vitamin E has the potential as an antioxidant to initiate wound healing due to sarcoma treatment. Vitamin E may also have protective effects on tumor tissue, mitigating the benefits of normal tissue protection. In a randomized trial, α-tocopherol was found to produce poor tumor control when used during the radiotherapy course [169,178].

#### 6.5.2. Melatonin

Melatonin, although not an antioxidant, can increase the expression of antioxidant enzymes such as SOD and glutathione peroxidase [179]. Several in vivo studies have established melatonin as a potent radioprotector [180,181,182]. Melatonin has also shown anti-tumor effects in in vivo models leading to tumor sensitization to RT [150,183]. In a phase II trial, melatonin was used to protect normal cells and sensitize tumor cells to RT [184]. Although melatonin was well tolerated, there was no improvement in long-term survival or neurologic function compared to controls [184]. Recently, Jin et al. showed melatonin induces wound healing in diabetic mice, suggesting a possible role in RT-associated impairment of wound healing [185].

#### 6.5.3. Growth Factors

Several growth factors are involved in normal tissue wound repair and are dysregulated after RT. Although these compounds are not typically considered radioprotectors, they can be thought of as radiation mitigators. Growth factors including TGF-β, VEGF, and PDGF appear to be important for mitigation of radiation-induced damage. VEGF was found to possess inhibitory effects on apoptotic cell death caused by ionizing radiation [186]. The topical application of PDGF has been shown to cure long-standing radiation-induced ulcers in vivo models [111,187]. TGF-β is considered crucial for cytokine involvement in the promotion of fibrosis by the deposition of excess collagen following radiotherapy [100]. RT causes the upregulation of TGF-β, as well as its receptor, TGF-βRII [188]. Inhibition of the effects of TGF-β by blocking its receptor has been found to reduce radiation fibrosis dramatically with fewer toxic effects [189]. Antisense TGF-β oligonucleotides showed an anti-scarring effect in an in vivo model of wound healing and scarring [190].

## 7. Conclusions

Neoadjuvant or adjuvant RT combined with surgical resection is a common standard of care treatment for STS, especially those that arise in the extremities; this combination allows for limb preservation and increased local control. Given the current rationale for RT in STS, the significant radiation dose to normal tissues is inevitable–regardless of RT modality or technique. Surgery in previously irradiated areas leads to more acute wound complications, whereas RT after surgery results in more late wound complications. Increasing the therapeutic window for any treatment modality requires simultaneously evaluating its effect on efficacy and toxicity. Ionizing radiation is an incredibly potent therapy for the malignant disease but is ultimately limited by the RT tolerances of normal tissues. Radioprotective agents ameliorate this issue but must be rigorously evaluated to ensure that they are not simultaneously decreasing the effectiveness of radiotherapy in malignant cells. Understanding wound physiology and identification of the cellular and molecular pathways affected by RT can lead to the development of novel therapeutics capable of both protecting normal tissues while enhancing the efficacy of radiotherapy in malignant cells-increasing the therapeutic index of radiotherapy from both sides of the equation. This provides a possible avenue for increasing local control while simultaneously decreasing acute and chronic toxicities and improving the quality of life in STS patients.

## Figures and Tables

**Figure 1 cancers-12-02258-f001:**
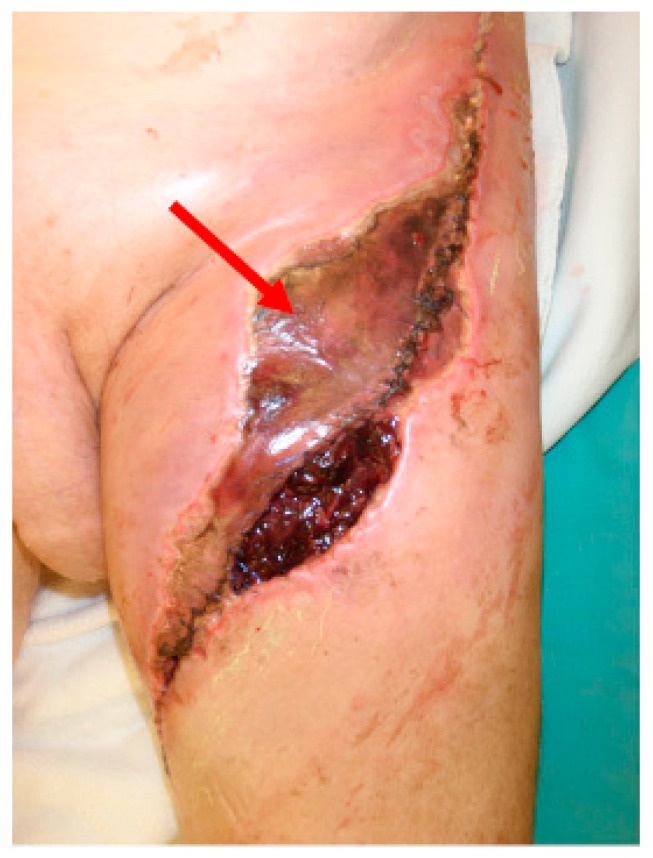
Clinical case of a patient treated with neoadjuvant radiotherapy in extremity soft tissue sarcoma. Wound necrosis (red arrow) was observed in the radiated area.

**Figure 2 cancers-12-02258-f002:**
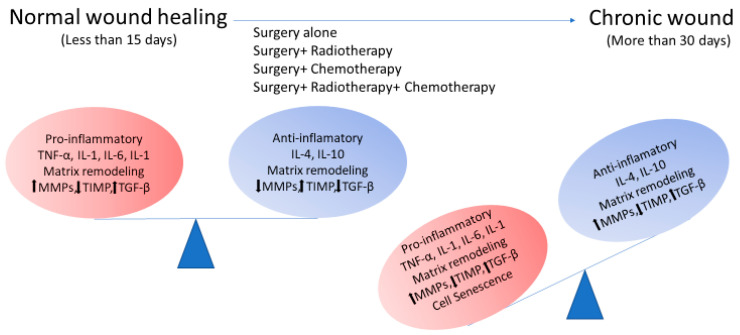
Chronic inflammation is the key feature in the radiogenic wound. In normal wound healing, there is a balance between the production of pro-inflammatory and anti-inflammatory cytokines, which is shifted towards a prolonged inflammatory phase in the radiogenic wound. Later, in the remodeling phase, there is an imbalance in the synthesis of matrix metalloproteinases (MMPs) and their tissue inhibitor (TIMP) in radiated skin.

**Figure 3 cancers-12-02258-f003:**
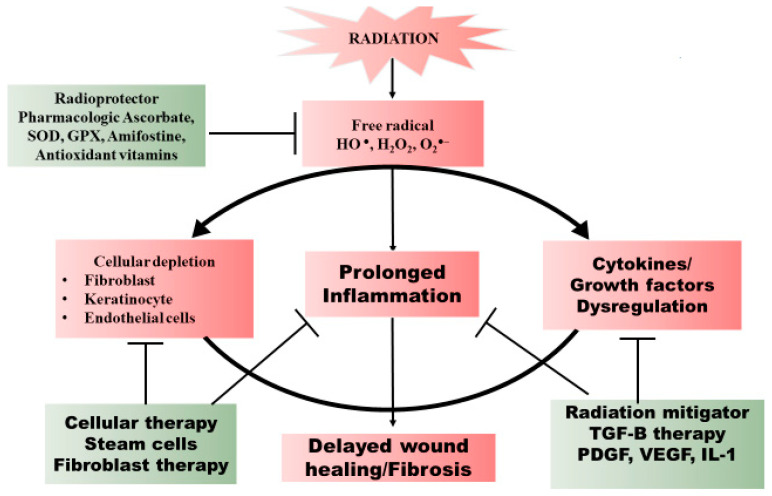
Summary of the key steps in wound healing dysregulated by radiation and the prospective therapeutic intervention of the events.

**Table 1 cancers-12-02258-t001:** Comparison of neoadjuvant vs. adjuvant acute and late wound complication in soft tissue sarcoma.

Reference	Disease Site	RT Course (# Patients)	Acute/ Late Toxicity	Measure	Neoadjuvant (%)	Adjuvant (%)	Significance
Pollack et al., 1998 [51]	MFH, synovial, and liposarcoma	Neoadjuvant 50Gy/25fx (*n* = 128), Adjuvant 60-66Gy/30–33fx (*n* = 165)	Acute	Wound complications	25%	6% *	*p* < 0.001
			Late	5-, 10-, and 15-year actuarial incidence	6, 7, and 7% respectively (Neoadjuvant & Adjuvant)		NS
O’Sullivan et al., 2002 [58]	Upper & Lower Extremities	Neoadjuvant 50Gy/25fx (*n* = 88), Adjuvant 66Gy/33fx (*n* = 94)	Acute	Skin toxicity grade ≥2	36%	68% *	*p* < 0.0001
				Wound complications	35%	17% *	*p* = 0.01
			Late	MSTS (mean, scale 0–35)	21	25 *	*p* = 0.01
				TESS (mean, scale 0–100)	60	69 *	*p* = 0.01
				SF-36 bodily pain (mean, scale 0–100)	58	67 *	*p* = 0.03
Zagars et al., 2003 [52]	Head & Neck, Trunk, and Extremities	Neoadjuvant 50Gy (*n* = 271), Adjuvant 60Gy (*n* = 246)(1.8–2.0Gy/fx)	Late	10-year actuarial complication incidence	5%	9% *	*p* = 0.03
				Necrosis, fractures, edema, or fibrosis	4%	8.9%	NR
Davis et al., 2005 [11]	Upper & Lower Extremities	Neoadjuvant 50Gy/25fx (*n* = 73), Adjuvant 66Gy/33fx (*n* = 56)	Late	Subcutaneous fibrosis	31.5%	48.2%	NS
				Joint stiffness	17.8%	23.2%	NS
				Edema	15.1%	23.2%	NS
				TESS (mean, scale 0–100)	85.1	81.3	NS
				MSTS (mean, scale 0–35)	29.9	28.0	NS
O’Sullivan et al., 2013 [30]	Lower Extremities	Neoadjuvant 50Gy/25fx (*n* = 59), Compared to historical control of neoadjuvant from Davis et al., 2005 [11]	Acute		O’Sullivan 2013	Davis et al., 2005	
				Secondary operation	33%	43%	NS
				Seroma/hematoma drainage	8.4%	NR	
				Infection requiring debridement	5.0%	NR	
				Dressing changes/deep packing. 4 months post-surgery	6.7%	NR	
				Total wound complications	30.5%	43%	NS
			Late	Edema	11.1%	15.1%	NR
				Skin Toxicity	1.9%	NR	
				Subcutaneous fibrosis	9.3%	31.5%	NR
				Fracture	0%	NR	
				Joint Stiffness	5.6%	17.8%	NR
				TESS (mean, scale 0–100)	83.1	85.1	NR
				MSTS-87 (mean, scale 0–35)	31.5	29.9	NR
				MSTS-93 (mean, scale 0–100)	89.3	NR	
Folkert et al., 2014 [36]	Upper & Lower Extremities	Neoadjuvant 50Gy median (*n* = 39), Adjuvant 63Gy median (*n* = 280)	Acute	Wound complications	17.5%	18.8%	NS
				Radiation dermatitis	48.7%	31.5%	*p* = 0.002
			Late	Fracture	9.1%	4.8%	NS
				Joint stiffness	11.0%	14.5%	NS
				Edema	14.9%	7.9% *	*p* = 0.05
				Nerve damage	1.6%	3.5%	NS
				Total	36.6%	30.7%	NR
Muller et al. 2016 [59]	Upper & Lower Extremities	Neoadjuvant 59Gy mean (*n* = 89), Adjuvant 71Gy mean (*n* = 365)	Acute	Surgical revision	9.0%	4.4%	NS
			Late	Wound necrosis, pathologic fractures, etc.	11.2%	15.2%	NS

Abbreviations: #= number, * = Significance at *p* < 0.05, fx = fractions, Gy = Gray, MFH = Malignant Fibrous Histiocytoma, MSTS = Musculoskeletal Tumor Society Rating Scale (with updates -87 and -93), NR = Not Reported, NS = Not Significant, SF = Short Form, & TESS = Toronto Extremity Salvage Score.

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
