# Peer review of "Neoadjuvant Radiotherapy-Related Wound Morbidity in Soft Tissue Sarcoma: Perspectives for Radioprotective Agents"

_cancers, 2020, doi:10.3390/cancers12082258_

Round 1

Reviewer 1 Report

The manuscript examined side effects of radiotherapy, discuss molecular mechanism of radiation damage and list preventive treatments.

The review is copletely narrative and qualitative. There is no added value to existing literature.

Paragraph 2.1 and 2.1.1 reports the same concepts and sometimes the same sentences.

The manuscript reports on some very basic aspects of radiotherapy and wound healing. This seems suitable for an educational book rather than a scientific article.

There are about 15 authors for a narratiev review. This is far too much for this kind of study. Paragraph repetition has not been picked up by anyopne of them questionioning their contribution.

Reviewer 2 Report

The current review article titled "Neoadjuvant Radiotherapy-related Wound Morbidity in Soft Tissue Sarcoma: Perspectives for Radioprotective Agents" presented the current literature about neoadjuvant/adjuvant radiotherapy and wound healing complications in soft tissue sarcoma, as well as novel radioprotective agents which could help reduce radiation-induced toxicity.

The review is generally well written and comprehensive. My main concern is that I do not clearly see any perspectives although this word is included in the title. I would therefore suggest that the authors provide some perspectives before or in the Conclusion.

In addition, the text should be thoroughly checked for language. For example:

  • Line 99: "by" should be deleted in "outcomes, and by disease site"
  • Line 110 "and but" does not make sense
  • The authors should make sure that they use "compared" appropriately: usually, compared with is used for contrast.

The following points should also be considered:

  • The announced structure "radiation modality, clinical regimen with associated toxicity outcomes, and disease site" is not perfectly respected, especially 2.2.
  • Table 1 is too busy and tough to read
  • Normal Physiology of Wound Healing: I have trouble fitting this section in the whole review. Since the related information is too simplified (especially 3.1 Homeostasis & Inflammation), I would suggest to reduce this section and use it as an introduction to 4. Wound Pathology After Radiation Treatment.
  • Support claims with references. See Lines 101-104. This is just an example.

Reviewer 3 Report

The authors are to be commended for their well written and comprehensive review of neoadjuvant radiation in soft tissue sarcoma.

A few minor points to consider:

  1. Rather than say recommendation “may also be used in cases with a concern for or (known) positive surgical margins”, I would suggest highlighting that radiation is recommended for microscopic or macroscopic residual disease.
  2. Even though the not the focus of the article, I would suggest some mention of the risk of secondary malignancies with radiation.
  3. Section 2.3.1 briefly touches on chemotherapy. I would suggest highlighting that chemotherapy is not standard of care in the management of soft tissue sarcomas but may be used in select circumstances for the less informed practitioner.
  4. Section 6.1 states amifostine is approved by ASCO For the prevention of xerostomia. Suggest that this be changed to is recommended by ASCO or is regulatory body approved (FDA).
